# Wastewater Treatment from Lead and Strontium by Potassium Polytitanates: Kinetic Analysis and Adsorption Mechanism

**Anna Ermolenko [1], Alexey Shevelev [1], Maria Vikulova [1], Tatyana Blagova [1], Sergey Altukhov [1], Alexander Gorokhovsky [1], Anna Godymchuk [2,3], Igor Burmistrov [1,4,5,*] and Peter Ogbuna Offor [6]**

[1] Department of Chemistry and Technology of Materials, Yuri Gagarin State Technical University of Saratov, 77 Polytecnicheskaya Street, 410054 Saratov, Russia; molish01@mail.ru (A.E.); titans5@rambler.ru (A.S.); vikulovama@yandex.ru (M.V.); blagova-ta@yandex.ru (T.B.); sergeypavlovicha@yandex.ru (S.A.); algo54@mail.ru (A.G.)

[2] Department of Materials Science, National Research Tomsk Polytechnic University, 30 Lenina Avenue, 634050 Tomsk, Russia; godymchuk@mail.ru

[3] Tobolsk Complex Scientific Station, Ural Branch of the Russian Academy of Science, 15 Osipov Academician Street, 626152 Tobolsk, Russia

[4] Department of Functional Nanosystems and High-Temperature Materials, National University of Science and Technology "MISiS", 4 Leninsky Avenue, 119049 Moscow, Russia

[5] Engineering Center, Plekhanov Russian University of Economics, 36 Stremyanny Lane, 117997 Moscow, Russia

[6] Metallurgical and Materials Engineering Department, University of Nigeria, Nsukka 410001, Nigeria; peter.offor@unn.edu.ng

[*] Correspondence: glas100@yandex.ru; Tel.: +7-917-201-87-03

**Abstract:** The reduction of heavy and radioactive metal pollution of industrial wastewater remains a vital challenge. Due to layered structure and developed surface, potassium polytitanate had potential in becoming an effective sorbent for metal extraction from wastewater in the presented paper. On the basis of the different sorption models, this paper studied the mechanism of $Pb^{2+}$ and $Sr^{2+}$ cation extraction from aqueous solution by non-crystalline potassium polytitanate produced by molten salt synthesis. The ion exchange during metal extraction from model solutions was proven by kinetic analysis of ion concentration change, electronic microscopy, and X-ray fluorescence analysis of sorbent before and after sorption, as well as by theoretical modeling of potassium, lead, and strontium polytitanates. The sorption was limited by the inner diffusion in the potassium polytitanate (PPT) interlayer space, as was shown using the Boyd diffusion model. The sorption processes can be described by Ho and McKay's pseudo-second-order model compared to the Lagergren pseudo-first-order model according to kinetic analysis. It was found that the ultimate sorption capacity of synthesized sorbent reached about 714.3 and 344.8 (ions mg/sorbent grams) for $Pb^{2+}$ and $Sr^{2+}$ ions, respectively, which was up to four times higher than sorption capacity of the well-known analogues. Therefore, the presented study showed that potassium polytitanate can be considered as a promising product for industry-scaled wastewater purification in practice.

**Keywords:** sorption; heavy metals; radioactive metals; potassium titanate; ion exchange

## 1. Introduction

Industrialization has caused the pollution of natural waters by industrial wastes containing heavy and radioactive metals from ore-beneficiation plants [1], metal treatment [2], paints and pigments [3],

and batteries [4]. Lead and radioactive strontium are toxic and cause environmental and human health damage [5]. Accumulation of heavy metals in the reservoirs closed by enterprises may bring severe hazards to the bio-system and society [6]; therefore, the reduction of the amount of metal in the wastewater remains a vital challenge.

Ion-exchange sorption is still one of the most effective and cost-beneficial methods for removing heavy metals from aqueous media [7–9], as it provides a high purification degree (up to 100%) of highly polluted waters, the possibility of sorbent regeneration, and simple use. The most common heavy metal adsorbents are synthetic zeolite, natural kaolinite, chitin, chitosan [10–14], carbon nano-tubes (CNT), carbon materials [15–17], agricultural waste such as rice bran, orange peel [18], montmorillonite, industrial by-products (such as lignin, diatomite, clino-pyrrhotite, lignite, aragonite shells, peat), bio-sorbents [19], and nanomaterials of metal oxides [20]. However, the introduction of most materials into the practice is limited because of its' high cost, sensitivity to occupational conditions, and complicated re-use. Therefore, the search for effective sorption materials is still important.

Potassium titanates have been considered as promising sorbents for heavy metals. In particular, sorption capacity of crystalline potassium hexatitanate ($K_2Ti_6O_{13}$) was found to reach 0.80 mmol/g in $Cd^{2+}$ solution [21]; meanwhile, the capacity of crystalline potassium tetratitanate ($K_2Ti_4O_9$) in $Cu^{2+}$ solution was 1.94 mmol/g [22]. The crystalline sodium titanate proved the capacity of 2.66 mmol/g compared to the 0.24 mmol/g for $H_2Ti_4O_9$ and 0.16 mmol/g for $H_2Ti_3O_7$ [22–24]. Besides crystalline potassium titanates, the special interest of researchers has been taken to potassium polytitanates (PPT). Due to the high surface area and the relatively large distance between the layers of the titanium–oxygen octahedron (1.5–2.0 nm), PPTs have increased ion-exchange capacity for heavy metal removal from wastewater [25,26]. Moreover, PPTs have a negatively charged surface, which can electro-statically attract metal oxide-hydroxide cations, as shown in our previous investigations of layered lepidocrocite-like quasi-amorphous compounds based on modified PPTs extracting organic dyes [27] and nickel ions [28].

Despite numerous studies on quantitative assessment of crystalline potassium titanate sorption properties, there is still a lack of knowledge about the sorption mechanism and ultimate PPT sorption capacity. Thus, the finding of this current study provides kinetic analysis to establish the mechanism of heavy metal ion sorption on a new and poorly studied type of X-ray amorphous sorbents—potassium polytitanates.

## 2. Materials and Methods

### 2.1. Synthesis and Characterization of PPT Sorbent

Potassium polytitanate (PPT) was synthesized by molten salt synthesis [25]. For the synthesis, the following chemicals were used for the mixture preparation: KOH (98% purity, Vekton, St. Petersburg, Russia), $KNO_3$ (98% purity, Reahim, Moscow, Russia), and $TiO_2$ (99% anatase, Aldrich, Darmstadt, Germany, CAS 13463-67-7). The dry mixture was stirred for 5 min in an alundum crucible with components mass ratio $KOH/KNO_3/TiO_2 = 30:40:30$ at $25 \pm 2$ °C. Then, after adding distilled water in the mass of twice the $TiO_2$ mass, it was stirred again for 5 min. Further, the mixture was heated with a heating rate of 7 °C/min up to $500 \pm 10$ °C and was held for 3 h in an electric muffle furnace PM-8 (TD Lab-Therm Ltd., Moscow, Russia) prior to natural cooling in the closed furnace for 24 h. The cooled product was triturated in an agate mortar and placed in a glass beaker filled with distilled water (powder/water ratio = 1:2) for thorough mixing. The resulting suspension was washed six times with distilled water by decanting until the pH of the washing water was $10.5 \pm 5$. The pH of the suspension was monitored with a 150MI pH meter (Measuring technology Ltd., Moscow, Russia). Then, the suspension was filtered and dried at $50 \pm 0.5$ °C in a Binder FD 53 drying oven (BINDER GmbH, Tuttlingen, Germany) for 24 h. The dry sample was ground into a powdered material in an agate mortar. Considering the material's hydrophilicity and the possibility of humidity change

during the long-term storage, all measurements were carried out for a freshly synthesized powder (no more than 1 week after synthesis).

The morphology and elemental analysis of the sorbent surface were performed with the use of a scanning electron microscope (SEM) with integrated energy-dispersive X-ray spectroscopy (EDX) analyzer EXplorer (ASPEX, Delmont, USA). After spraying gold onto a sorbent placed on an aluminum substrate, measurements were carried out at the accelerating voltage of 15 kV. The individual particles of the synthesized titanates were studied in terms of morphology and size by a transmission electron microscope (TEM) JEOL JEM 1400 with an accelerating voltage of 120 kV (JEOL Ltd., Tokyo, Japan). The PPT-specific surface was determined with the Quantachrome Nova 2200 analyzer (Quantachrome Instruments, Boynton Beach, USA) by using the initial part of the isotherm of physical sorption of high pure nitrogen (99.999%) and was calculated by instrumental software using the Brunauer-Emmett-Teller (BET) model. Preliminary degassing of the samples was carried out at 150 ± 2 °C for 3 h. Own instrumental error did not exceed 2%.

The phase composition of the synthesized sorbent was determined using an X-ray diffractometer ARL X'TRA (Thermo Scientific, Ecublens, Switzerland) with CuKα radiation (λCuKα = 0.15412 nm) in a 2Θ angle range from 5 to 60 degrees. Bragg–Brentano measurement geometry was used for analysis with step-by-step scanning mode, speed of 2 degrees, and signal accumulation time of 1 s. The library of the international electronic database of diffraction standards (produced by ICDD—International Center for Diffraction Data)—PDF-2 (Powder Diffraction File-2) database in the Crystallographic Search-Match Version 3,1,0,2 B was used for phase identification on the resulting diffractograms. The permissible absolute error limits in measuring the angular positions of the diffraction maximum were ±0.0015°. Particle-size distributions of the ceramic filler and polymer particles in the dispersion were obtained using Analysette 22 MicroTec Plus Fritsch laser diffraction equipment.

The performed quantum modeling of the PPT structure facilitated the understanding of the material structure and the explanation of possible sorption mechanisms. Thus, optimizing to minimum potential energy, the clusters $K_4Ti_8O_{18}$, $Pb_2Ti_8O_{18}$, and $Sr_2Ti_8O_{18}$ were modeled. As a modeling result, the calculated "cation-oxygen" distances in the PPT structure were obtained.

A quantum chemical study was performed by the Priroda 6 program, using the density functional theory (DFT), Perdew–Burke–Ernzerhof (PBE) functional [29], L1 basis, using scalar relativistic corrections [30,31] prior to the calculation of the cation-oxygen inter-atomic distances in the compounds. The obtained values allowed for the creation of the PPT cluster by incrementally changing the geometry of the molecules with optimization of the system potential energy to the minimum.

## 2.2. Sorption Capacity Assessment

Lead nitrate ($Pb(NO_3)_2$, 99.5% pure, Vekton Ltd., Russia) and strontium 6-water chloride ($SrCl_2 \cdot 6H_2O$, CAS 10025-70-4 Vekton Ltd., Russia) were used to prepare two separate aqueous solutions (for a separate measurement of lead and strontium ions sorption) with $Pb^{2+}$ or $Sr^{2+}$ concentration of 50 mg/mL (hereinafter in the calculations—$C_0$). The solutions were prepared on the basis of distilled water (pH = 6.0–6.5) right before the experiment. The pH of the initial solutions were 4.50 ± 0.20 and 10.65 ± 0.30, respectively, for $Pb^{2+}$ and $Sr^{2+}$ solutions. The PPT was exposed to the prepared solution (sorbent concentration 110 g/L) at 25 ± 0.2 °C for 90 min with constant stirring using a magnetic stirrer (60 rpm). Throughout the experiment, the pH of the system was measured using an MA130 ion-meter (Mettler-Toledo GmbH, Greifensee, Switzerland).

Every 5 min, 5 mL of the suspension was taken and filtered on a Bola GmbH PTFE filter (Bohlender GmbH, Grünsfeld, Germany, pore size—10 μm, pressure—50 mm Hg). The concentration of metal ions (ion concentration at the given point in time, $C_t$) was determined in the filtrate using the X-ray fluorescence method on the Spectroscan-MAX-G spectrometer (Spectron, Moscow, Russia) with a scanning crystal diffraction channel. The measurements were carried out at 20 °C using an X-ray tube with a silver anode (4 W, 40 kV, and 100 μA) and LiF crystal analyzer (200) in the interval—810...3200 mA with a scanning step of 2.0 mA. The method sensitivity for metal ions was 1–20 ppm. Kinetic curves

(($\Delta C = f(t)$, where $\Delta C = C_0 - C_t$ is the change in concentration by the time t, min) were created on the basis of obtained data. Reaching the constant $\Delta C$ value means the occurrence of the dynamic equilibrium corresponding to the metal ions concentration $C_1$. The sorption capacity $Q$ (mg of metal/g of sorbent) was calculated by Equation (1):

$$Q = \frac{(C_0 - C_t)V}{m} \tag{1}$$

where $C_0$ is the initial metal ion concentration, g/L; $C_t$ is the metal ion concentration after the sorbent/sorbate dynamic equilibrium onset, g/L; $V$ is the volume of the metal salt solution, l; and m is the mass of sorbent, g.

### 2.3. Numerical Analysis of Adsorption Kinetics

Sorption kinetics analysis was performed by modeling after a detailed comparison of several key models to evaluate the contribution of various sorption mechanisms. Boyd's diffusion model is currently the only kinetic model allowing simultaneous examination of external and internal diffusion as limiting sorption stages. This approach simplifies the analysis by eliminating the need for multiple recalculations of experimental data using several models.

By the Boyd model, external diffusion processes were characterized by the time dependence $-\log(1-F)$, where $F$ is the equilibrium degree in the system calculated by Equation (2):

$$F = \frac{Q_t}{Q_e}, \tag{2}$$

where $Q$ is the adsorbed substance amount per sorbent mass unit at the moment of time t ($Q_t$) and in the equilibrium state ($Q_e$, mg/g).

The sorption limitation by intra-diffusion processes was determined by the $F - Bt$ dependence, where $Bt$ is the dimensionless Boyd parameter calculated more accurately by Equation (3):

$$Bt = 0.4977 - \ln(1 - F) \tag{3}$$

The identification of the influence of the chemical stage of ion exchange on the process of PPT interaction with metal ions was performed considering sorption using the kinetic model of pseudo-first and pseudo-second Lagergren's orders, which takes into account the solid sorbent sorption capacity [32], described by the Equation (4):

$$\log(Q_e - Q_t) = logQ_e - \frac{kt}{2.303} \tag{4}$$

where $k$ is the sorption rate constant, $min^{-1}$.

The correspondence of the obtained experimental data of the pseudo-second-order model was performed using the Ho and McKay classical equation [33] in the linear form:

$$\frac{t}{Q_t} = \frac{1}{k_2 Q_e^2} + \frac{t}{Q_e} \tag{5}$$

where $k_2$ is the rate constant, $g \cdot mmol^{-1} \cdot min^{-1}$.

## 3. Results and Discussion

### 3.1. Characterization of Sorbents

The morphology, structure, and elemental and phase composition of PPT was studied before and after sorption. The synthesized PPT was a powder with particle size distribution from 100 nm to 10 μm (Figure 1a,b). Large particles were mainly aggregates of individual particles, having a layered structure and consisting of flat flakes with sizes mainly from 300 to 1000 nm and thickness of several

atomic layers (Figure 1c). The specific surface area of the obtained PPT was from 90 to 100 m$^2$/g (variation in the series of experiments). X-ray diffractogram of the PPT before sorption (Figure 2b) did not include specific peaks, whereas the wide reflex at 48 degrees indicated the formation of the PPT structure [25,34]. The broad peak, amorphous halo, in the range between 25 and 35 degrees of 2Θ testified the X-ray amorphous structure of PPTs and the absence of long-range order in the lattice periodicity.

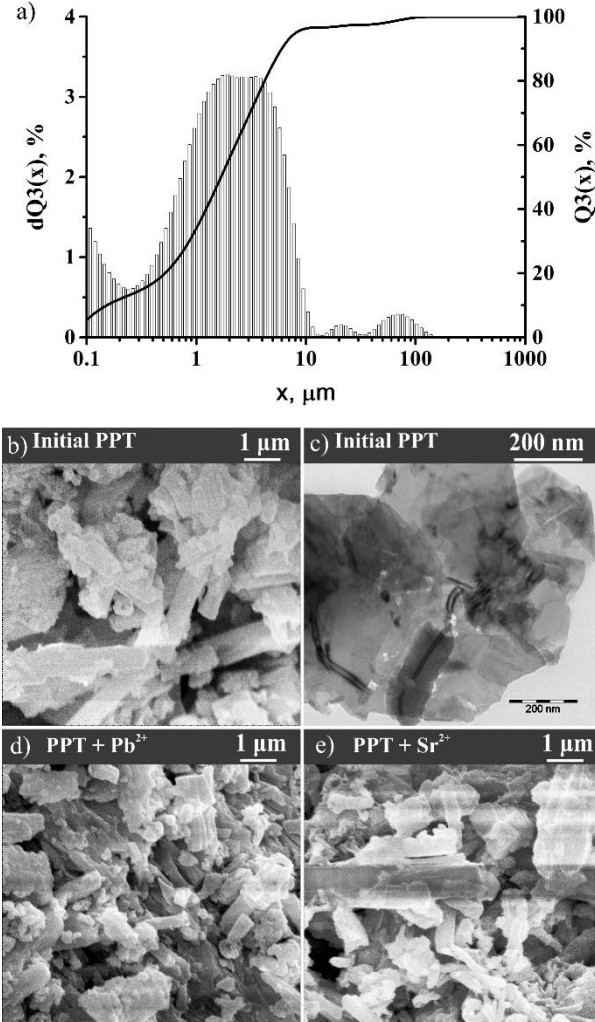

**Figure 1.** Particle size distribution of potassium polytitanates (PPT) before sorption (**a**); scanning electron microscope (SEM) (**b**) and transmission electron microscope (TEM) image (**c**) of PPT before sorption, and SEM image of PPT after sorption of Pb$^{2+}$ (**d**) and Sr$^{3+}$ (**e**).

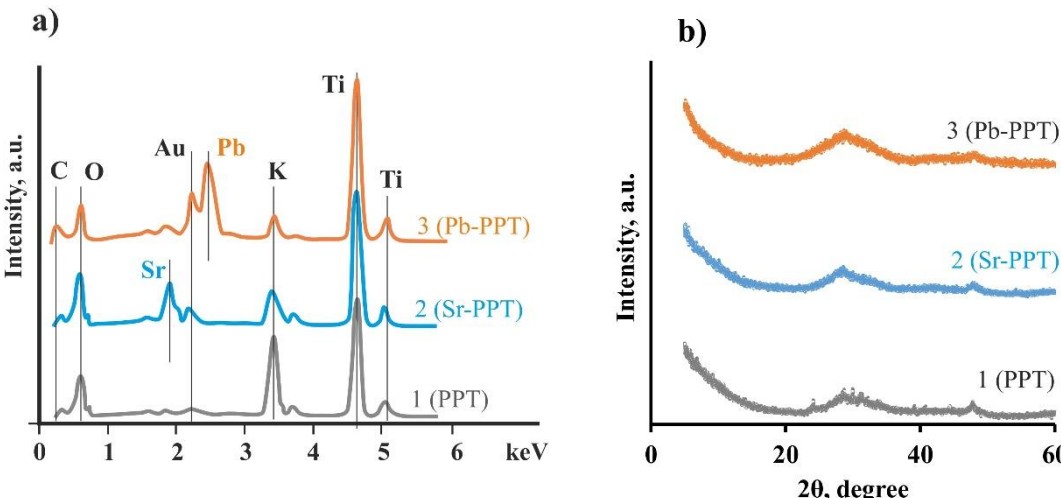

**Figure 2.** Chemical composition data (energy-dispersive X-ray spectroscopy (EDX)) (**a**) and X-ray (**b**) of synthesized PPT and exposed sorbents (Sr-PPT and Pb-PPT).

It was shown that the particle morphology of exposed sorbents did not differ from the initial form (Figure 1d,e). The adsorption intensity of the K band decreased, whereas Pb and Sr bands appeared according to its X-ray fluorescence spectrum (Figure 2a). The obtained data suggest the ion exchange of $K^+$ ions with $Pb^{2+}$ and $Sr^{2+}$ on the PPT surface. In the series of $K^+$, $Pb^{2+}$, and $Sr^{2+}$ ions, the bond length between cat-ion and oxygen, calculated using the density functional theory, decreased in the structure of the PPT (Table 1). The obtained values of the bond length were relatively close to those found in the literature data [35]. The discrepancy with the literature data was due to the limited size of the calculated cluster and estimated errors of the optimization of the system potential energy to the minimum.

**Table 1.** Inter-atomic distance "cation-oxygen" for $K_4Ti_8O_{18}$ and $Me_2Ti_8O_{18}$ (Me = Pb or Sr).

| Complex | Cation | Calculated Distance "Cation-Oxygen", nm | Reviewed Distance "Cation-Oxygen" [35], nm | Divergence with Literary Values, % |
|---|---|---|---|---|
| $K_4Ti_8O_{18}$ | $K^+$ | 0.280 | 0.269 | 4 |
| $Pb_2Ti_8O_{18}$ | $Pb^{2+}$ | 0.232 | 0.262 | 11 |
| $Sr_2Ti_8O_{18}$ | $Sr^{2+}$ | 0.261 | 0.256 | 2 |

On the basis of the smaller inter-atomic distance indicating a stronger bond of $Pb^{2+}$ and $Sr^{2+}$ ions with oxygen compared to $K^+$, it can be assumed that the strong chemical connection of $Pb^{2+}$ and $Sr^{2+}$ ions would minimize the probability of desorption of ions and, therefore, prevent secondary environmental pollution.

*3.2. Change of Sorption Rate*

Figure 3 shows the time- and pH-dependent kinetics and limited sorption capacity of $Pb^{2+}$ and $Sr^{2+}$ ions. According to the kinetic curve of $Pb^{2+}$sorption, the induction period accompanied the sorbent's aging during the first 5 min of being on (Figure 3a, $Pb^{2+}$-ion curve). This effect may have been caused by the slow destruction of inert lead complexes followed by its accumulation near the surface, which is typical of transition metal solutions [36,37]. Then, the lead sorption in the solution started to intensify, and in 30 min the system achieved a quasi-equilibrium state at $\Delta C = 42.0 \pm 1.9$ mg/mL. The second equilibrium occurred at 45 min, when the efficiency of $Pb^{2+}$ ion extraction was 99.4% (the maximum decrease in concentration was $49.7 \pm 1.5$ mg/mL). The complex mechanism of PPT

interaction with $Pb^{2+}$ ions was caused by the layered structure of the sorbent—the first equilibrium at the initial stage could be most likely be explained by the ion adsorption on the PPT surface, whereas the second equilibrium was related to sorbate diffusion into the PPT interlayer space. Because the pH of the $Pb^{2+}$-based PPT suspension during the sorption process was within the range of 4.5–6.2 (Figure 3b), sedimentation of oxide–hydroxide metal complexes was not expected, and $Pb^{2+}$ ion intercalation processes were very possible.

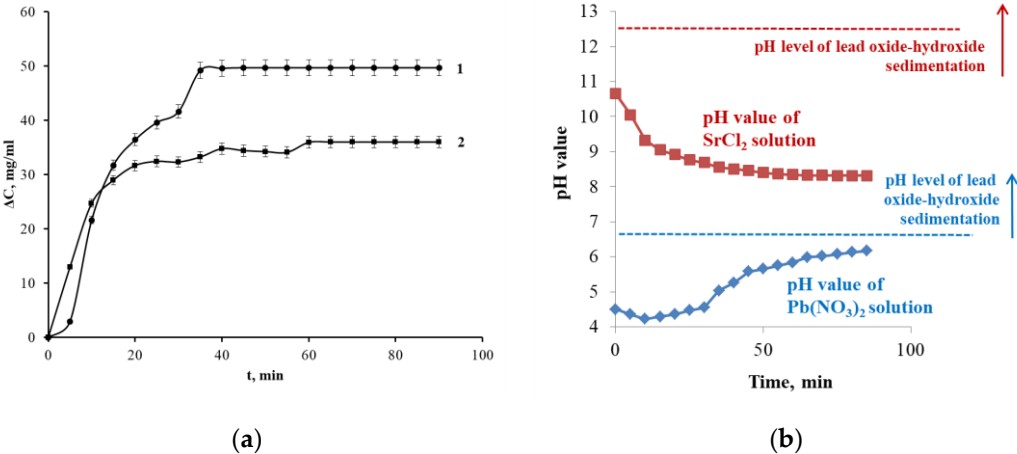

**Figure 3.** Sorption kinetic curve of $Pb^{2+}$ (1) and $Sr^{2+}$ (2) ions on PPT at pH < 6.5 and pH > 8.5, respectively (**a**), and the pH value during the $Pb^{2+}$ and $Sr^{2+}$ sorption on PPT (**b**).

When extracting $Sr^{2+}$ ions in the first 10 min, the concentration of the ion changed linearly to $\Delta C$ = 25.0 ± 1.6 mg/mL (Figure 3, $Sr^{2+}$ ion curve). Then, it increased gradually, and after 45 min reached a maximum value of 36.0 ± 1.0 mg/mL; at that point, the degree of extraction reached 72%. This kinetic curve behavior was likely due to the twice larger ionic radius of $Sr^{2+}$ than $Pb^{2+}$, complicating the diffusion of the ion into the PPT interlayer space. In this regard, there was a single equilibrium state related to the ion sorption on the sorbent surface. Moreover, the pH of the $Sr^{2+}$-based PPT suspension during the sorption process exceeded 8.5 (Figure 3b), meaning that the basic medium contributed to the formation of surface oxide–hydroxide metal complexes, and excluded the intercalation process.

However, the kinetic analysis did not explain the differences in sorption of cations, and it did not reveal the mechanism of sorption combining the transfer of cat-ions and chemical interaction of PPT with metal ions.

### 3.3. Analysis of the Diffusion Component of the Sorption Process (Boyd Model)

To assess the role of cations transfer to and inside the solid surface, we used the diffusion model (Boyd model). Within the Boyd diffusion models, a quantitative approach was applied for the primary distinction between intra- and external-diffusion adsorption limitation, which involved the analysis of kinetic data in the coordinates $-\ln(1-F)$ and $t$.

According to the Boyd model, the linear dependence in the coordinates $(-\ln(1-F))$-t indicates the external diffusion of ions to and along the surface of the sorbent as the limiting stage of sorption. The coefficient of approximation accuracy for given dependences was relatively low over the entire time range for studied systems: $R^2 \approx 0.9$ for $Pb^{2+}$ and $R^2 < 0.9$ for $Sr^{2+}$ ions (Figure 4). However, the identification of the linear areas for both metal ions became possible within the first 25 min of interaction. Therefore, the $Pb^{2+}$ sorption was limited by external (film) diffusion only within first contact stages (sorption time was less than 25 min). Then, while the PPT active centers were being filled, the impact of intra-diffusion mass transfer on the sorption process increased. The logarithmic dependence of the equilibrium attainment degree on the Boyd parameter (Figure 5) obtained for the sorption of $Pb^{2+}$ ions proved the intra-diffusion mechanism of sorption. The high approximation

confidence coefficient of the given curve of the logarithmic trend line ($R^2 > 0.98$) indicated a large contribution of the internal diffusion of metal ions in the sorbent interlayer space to the overall rate of the sorption process.

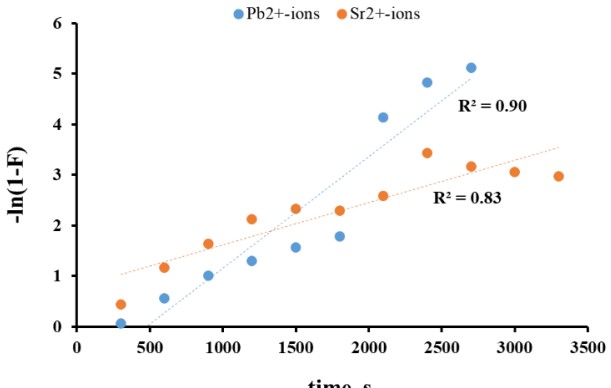

**Figure 4.** Change of $-\ln(1-F)$ coefficient for metal ion sorption on PPT.

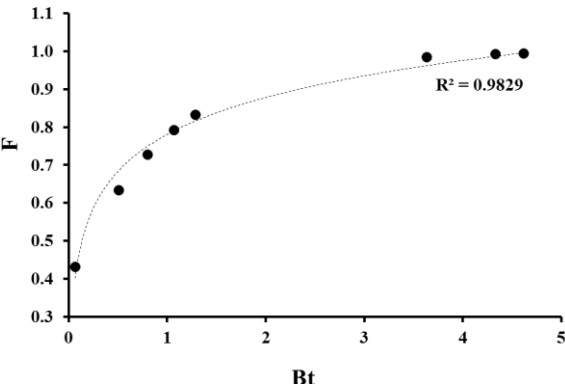

**Figure 5.** Change of equilibrium attainment degree (F) depending on Bt during $Pb^{2+}$ ion sorption.

Analysis of the data from Figures 3–5 suggested a mixed kinetic mechanism (including components of external and internal diffusion) for the $Pb^{2+}$ ion sorption at PPT, where the sorption process was limited by the stage of external diffusion of ions to and along the sorbent surface at the beginning of the interaction between metal ions and PPT, and then internal diffusion in the PPT interlayers became the slowest stage after reaching the certain saturation degree of sorption centers. Thus, the kinetic curve had two equilibrium parts related to adsorption and intercalation processes. However, the $Sr^{2+}$ ion sorption rate at the first stage of interaction with PPT was limited only by the external diffusion of ions in the solution and on the surface of the sorbent (Figure 4) due to discrepancy of $F - Bt$ dependencies for the $Sr^{2+}$ solution, proving insignificant impact of internal diffusion on the final sorption rate.

The use of the Boyd model showed a complex interaction of PPTs with $Pb^{2+}$ ions, and the limiting stage of external and internal diffusion was difficult discover because the total sorption rate was determined by a different mechanism of diffusion transfer of metal cations.

### 3.4. Analysis of Chemical Kinetics of Sorption Processes

The kinetic models of pseudo-first and pseudo-second orders can allow for the estimation of the contribution of chemical interaction of sorbent and cation into the sorption kinetics. When analyzing the sorption process using a pseudo-first-order model (Figure 6), the linear dependencies of $\log(Q_e - Q_t)$ with a sufficiently low coefficient of $R^2$ were obtained. Furthermore, the $R^2$ for $Sr^{2+}$ ions (Figure 6, $Sr^{2+}$ ion curve) lay below $R^2$ for $Pb^{2+}$ ions (Figure 6, $Pb^{2+}$ ion curve). The results of the experimental data treatment indicated the diffusion presence as being a preliminary stage in the sorption process of ions

on PPT, though its role was small, especially in the case of $Sr^{2+}$ ions. Thus, following the Lagergren model, the sorption of $Pb^{2+}$ and $Sr^{2+}$ ions was limited by the chemical ion exchange reaction stage, whereas the diffusion stage preceding ion-exchange sorption had a small additional impact on the reaction between sorbate and functional group of the sorbent.

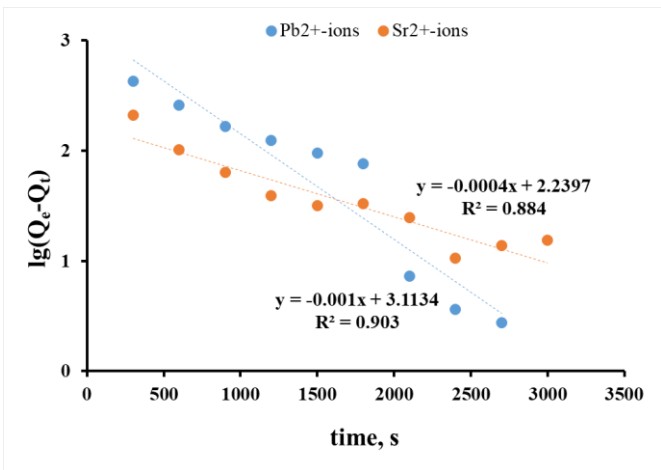

**Figure 6.** Sorption kinetics of $Pb^{2+}$ and $Sr^{2+}$ ions in the coordinates of the Lagergren pseudo-first-order.

The ongoing sorption processes more likely corresponded to the pseudo-second-order of Ho and McKay's model (Figure 7), regardless of the ion nature. The value of $R^2$ for the $\log(Q_e - Q_t)$ dependencies was 0.903 and 0.884 for $Pb^{2+}$ and $Sr^{2+}$ ions, respectively (Figure 6), compared to 0.971 and 0.997 for the $t/Q_t$ dependence (Figure 7).

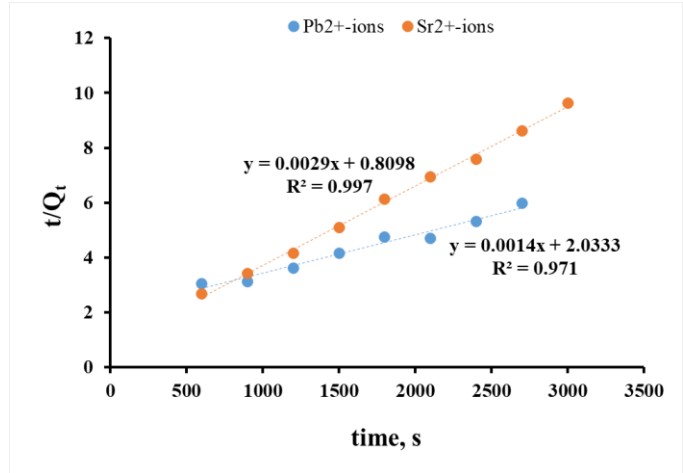

**Figure 7.** Sorption kinetics of $Pb^{2+}$ and $Sr^{2+}$ ions in the coordinates of the Ho and McKay pseudo-second-order.

Linear character of the dependences in the pseudo-second-order model's coordinates proved the limitation of ion sorption by the ion exchange stage; herein, this dependence was more defined for $Sr^{2+}$, and the reaction between the sorbent and the sorbate was a second-order reaction (they interacted with each other in one equivalent ratio). The linear curves allowed for the definition of the capacity and rate constant of the sorption (Table 2).

**Table 2.** Sorption kinetic parameters of $Pb^{2+}$ and $Sr^{2+}$ ions and PPT sorption capacity.

| Ions | Rate Constants | | PPT Sorption Capacity for Pseudo-Second-Order Model |
|---|---|---|---|
| | k, $min^{-1}$ | k, $g \cdot mg^{-1} \cdot min^{-1}$ | |
| | Pseudo−First Order Model | Pseudo−Second Order Model | |
| $Pb^{2+}$ | 0.0023 | $9.6 \cdot 10^{-7}$ | 714.3 |
| $Sr^{2+}$ | 0.0009 | $10 \cdot 10^{-6}$ | 344.8 |

Analysis of the sorption kinetic parameters, namely, the pseudo-first- and pseudo-second-order sorption constants, as well as the sorption capacity, allowed us to find out the multistage interaction mechanism of PPT with metal ions. This was most likely explained by the large PPT specific surface area (90–100 $m^2$/g), as well as sufficiently large interlayer distance, which both caused the interaction of the sorbate on the outer surface with thiol groups (-TiOH) and on the inner surface by ionic substitution of potassium cations.

The pseudo-first-order model described indeed chemical sorption, as most probable chemical reactions including ion exchange and covalent bonding are confirmed by good compliance of experimental data. Herein, the assumption made when the chemical composition analysis of the PPT surface before and after sorption was proven by the ion exchange as a limiting stage of the metal cation absorption by PPT.

*3.5. PPT Sorption Capacity Compared to Published Sorbents*

On the basis of the calculated sorption capacity Q (mg of metal/g of sorbent) (Table 2) and the experimentally determined equilibrium time $\tau_{eq}$ (45 min, Figure 3), PPT sorption capacity exceeded the capacity of the most known titanium sorbents (Table 3).

**Table 3.** Reported adsorption capacities towards heavy metal ions by typical adsorbents.

| Adsorbent | Ion | Adsorption Capacity, mg/g | Equilibrium Time, min | Adsorption Conditions | Reference |
|---|---|---|---|---|---|
| PPT | $Pb^{2+}$ | 714.3 | 45 | pH = 4.5–6.2 | This article |
| | $Sr^{2+}$ | 344.8 | | PH = 8.5–10.8 | |
| Chitosan/TiO$_2$ hybrid adsorbent | $Pb^{2+}$ | 36.8 | | 60 °C pH = 3–4 | [38] |
| $Na_2Ti_3O_7$ | $Pb^{2+}$ | 563.6 | 60–120 | - | [39] |
| Titanates with various morphology | $Pb^{2+}$ | 105–304 | 90 | - | [40] |
| Magnetic titanium nanotubes | $Pb^{2+}$ | 442.5 | 60 | pH = 5 | [41] |
| Titanate nanofibers | $Pb^{2+}$ | 244–280 | - | pH = 6–7 | [42] |
| | $Sr^{2+}$ | 50–55 | - | | |
| Titanate nanotubes | $Pb^{2+}$ | 2.6 | - | pH = 5 | [43] |
| | | 299.5 | - | | [44] |
| Titanate nanotubes obtained by hydrothermal method | $Pb^{2+}$ | up to 2000 | 30 | pH = 4 | [45] |
| | | 520.8 | 180 | pH = 5–6 | [46] |
| | $Sr^{2+}$ | 91.7 | 10 | - | [47] |
| | | 98.7 | 30 | - | [48] |

Thus, the sorption capacity of the synthesized PPT can exceed the capacity of most known literature titanium-containing sorbents with developed morphology of 1.3–270 times for $Pb^{2+}$ and 3.5 times for $Sr^{2+}$.

## 4. Conclusions

In this study, X-ray amorphous potassium polytitanate sorbent with a layered structure and well-developed outer and inner surfaces were synthesized by a molten salt method, and its sorption characteristics were investigated. The sorption equilibrium during PPT aging in $Pb^{2+}$ (pH< 5.5) and $Sr^{2+}$ (pH< 8.5) solutions with a concentration of 50 g/L occurred in 45 min. The sorption capacity of PPT was calculated on the basis of a kinetic parameter study designed in accordance with pseudo-second order model and was equal to 714.3 mg/g and 344.8 mg/g for $Pb^{2+}$ and $Sr^{2+}$ ions, respectively.

The multi-stage interaction mechanism between ions and sorbent, including outer diffusion to sorbent surface, inner diffusion on sorbent surface, inner diffusion in the PPT interlayer space, and ion exchange chemical reaction where ion exchange occurs as a liming stage, was developed using Boyd's diffusion model and pseudo-first- and pseudo-second-order chemical kinetic models. Herein, $Pb^{2+}$ sorption rate was significantly affected by the outer (within first minutes of interaction) and inner diffusion. Energy-dispersive analysis and quantum chemical study proved the lead and strontium presence in the PPT composition after sorption.

In summary, the sorption capacity of potassium polytitanate sorbent obtained in this study was significantly higher than the capacity of the most ceramic materials synthesized previously [38–47]. Despite better sorption of metal cation on carbon-based sorbents [15–17], ceramic materials are more preferable because of their re-use for various practical applications. For this reason, the synthesized material is a promising product for industry-scaled wastewater purification in practice.

**Author Contributions:** A.E.—Investigation; A.S.—Formal analysis; M.V.—Writing—original draft; T.B.—Visualization; S.A.—Validation; A.G. (Alexander Gorokhovsky)—Conceptualization; A.G. (Anna Godymchuk)—Writing—review & editing; I.B.—Project administration; P.O.O.—Writing—review & editing. All authors have read and agreed to the published version of the manuscript.

**Funding:** This research received no external funding.

**Conflicts of Interest:** The authors declare no conflict of interest.

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
