# Peer review of "Wastewater Treatment from Lead and Strontium by Potassium Polytitanates: Kinetic Analysis and Adsorption Mechanism"

_processes, doi:10.3390/pr8020217_

Round 1

Reviewer 1 Report

Review processes-690198

Title: Wastewater treatment from lead and strontium by potassium polytitanates: kinetic analysis and adsorption mechanism

Comments from reviewer

I have some comments on this manuscript.

In the paper, there are too many parenthesis used, so I suggest you to either remove redundant / unnecessary parenthesis or to simply them. It makes readers hard to capture the contents of the paper. In the introduction, there are too many technical term, especially, on line 52, I don’t think you’ll need to describe about the formula in this section. Content-wise, I think authors provides explanation and illustration of most key ideas, but missing incorporating some key concepts of the discipline. Also, the paper analyzed differing ideas and tried to synthesize all materials, but I also see incomplete synthesis of materials. In terms of organization, I’m concerning that authors provides inconsistent and sometimes unclear logic and arrangement of ideas. Also, I’m seeing minimal uses of transitions. Please carefully go / read over the manuscript again and revise the organization of the paper between sentences and paragraphs in order to make efficient transitions and to provide highly logical and clear arrangement of ideas. In terms of English language, I think authors writes some grammatically incorrect sentences, and has some comma splices, run-ons, and/or fragments but not a clear pattern in any error type; some misspellings

Author Response

Thanks so much for reviewing our article and such valuable comments!

(1). In the paper, there are too many parenthesis used, so I suggest you to either remove redundant / unnecessary parenthesis or to simply them. It makes readers hard to capture the contents of the paper.

Answer. Thank you for this valuable comment. We removed several parenthesis or simplified them and highlighted the revised phrases pink color.

(2) In the introduction, there are too many technical term, especially, on line 52,

Answer. Thank you for this comment. This is true that we used specific terms to chemical compounds and methods for their characterization. We tried to make the text more versatile and understandable. We also added some explanation for specific terms as “… Potassium titanates have been considered as a promising sorbents for heavy metals. In particular” and “Besides crystalline potassium titanates, the special interest of researchers has been taken to potassium polytitanates (PPT). Due to the high surface area and the relatively large distance between the layers of the titanium-oxygen octahedron”on the page 2 and “… functional theory (DFT), Perdew–Burke-Ernzerhof (PBE) functiona …l” on the page 3.

We deleted redundant titanate characterization as the text “Potassium titanates with formula K2O · nTiO2 (n = 2 ... 8) are functional materials which structure and properties are determined by the value of n. The structure includes negatively charged titanium-oxygen pentahedrons or octahedrons forming layered or tunnel structures, and potassium cations compensating the charge, located both on the surface and in the interlayer space (in channels in case of tunnel structures). Potassium cations have an ability to exchange with various inorganic and organic ions, which ensures the ion-exchange properties of titanates.” from the page 2.

(3) I don’t think you’ll need to describe about the formula in this section.

Answer. This is a general formula for an investigated substance. On the one hand, this formula is necessary to understand the object of research - the filler, on the other hand, it is well known, and we placed it in the introduction on the page 2.

(4) Content-wise, I think authors provides explanation and illustration of most key ideas, but missing incorporating some key concepts of the discipline. Also, the paper analyzed differing ideas and tried to synthesize all materials, but I also see incomplete synthesis of materials.

Answer. Thank you for paying attention to this, the section “Materials and Methods” has been improved accordingly: information about particle size distribution has been added. Kindly see the Figure 1 on the page 5 (line 178) and the text “Particle-size distributions of the ceramic filler and polymer particles in the dispersion were obtained using Analysette 22 MicroTec Plus Fritsch laser diffraction equipment” on the page 3.

(5) In terms of organization, I’m concerning that authors provides inconsistent and sometimes unclear logic and arrangement of ideas. Also, I’m seeing minimal uses of transitions. Please carefully go / read over the manuscript again and revise the organization of the paper between sentences and paragraphs in order to make efficient transitions and to provide highly logical and clear arrangement of ideas.

Answer. Thank you for this valuable comment, since there is lack of efficient transitions and logical and clear arrangement of ideas. After our additions the paper improved. We restructure the section “3. Results and discussion” changes the subsections 3.1-3.4 added the text “The morphology, structure, elemental and phase composition of PPT was studied before and after sorption.” on the page 4, “Figure 3 shows the time- and pH-dependent kinetics and limited sorption capacity of Pb2+ and Sr2+ ions.” on the page 6, “However, the kinetic analysis does not explain the differences in sorption of cat-ions, and it does not reveal the mechanism of sorption combining the transfer of cat-ions and chemical interaction of PPT with metal ions.” and “To assess the role of cations transfer to and inside the solid surface, we used the diffusion model (Boyd model).” on the page 7, “The use of the Boyd model shows complex interaction of PPTs with Pb2+ ions, and the limiting stage of external and internal diffusion is difficult to be revealed since the total sorption rate is determined by a different mechanism of diffusion transfer of metal cations.” And “The kinetic models of pseudo-first and pseudo-second orders can allow estimating the contribution of chemical interaction of sorbent and cation into the sorption kinetics.” on the page 7, on the page 7, on the page 7,  

(6) In terms of English language, I think authors writes some grammatically incorrect sentences, and has some comma splices, run-ons, and/or fragments but not a clear pattern in any error type; some misspellings

Answer. We appreciate you careful attention to the paper. The language of the paper was edited. All revisions were highlighted with pink color.

Reviewer 2 Report

This manuscript describes the adsorption capacity of PPT in extracting lead and strontium from aqueous waste. The material described does a very good job of extracting the pollutants, however the manuscript needs a considerable amount of work before it is fit for publication, including several comments I have outlined before. In general, the written expression is poor and impedes clear communication of the results. My largest concern is that the characterization of the PPT does not seem sound, and this undermines confidence in the manuscript as a whole.

1. Are lead and strontium being adsorbed simultaneously? This is not clear

2. "the pH of the suspension was controlled with a 150MI pH meter"  - controlled or monitored?

3. Some confusion in describing Figure 2 - a and b seem to be mixed in the text.

4. XRD looks very different to cited papers - what does “the wide reflex” mean? The assertion you make here is not supported by the data.

5. Was there any verification of the ratio of the components (using EDX?)

6. Table 1: is there an understanding of uncertainty here. Divergence with literature values unexplained. What is calculated vs reviewed?

7. Results of using models are not well explained and many citations are missing. Explain the models used better and what they mean in terms of physical understanding

8.…s throughout in numbers - are they meant to be emdashes?

9. Section 3.5 needs a table

10. Template text left in at the end of the Results and Discussion section!

Author Response

Reviewer#2.

Thanks so much for reviewing our article and such valuable comments!

(1) My largest concern is that the characterization of the PPT does not seem sound, and this undermines confidence in the manuscript as a whole.

Answer. Thank you for paying attention to the deficiency in method. We added the detailed characterization of PPT. Kindly see the “The broad peak, amorphous halo, in the range of 25…35 degrees of 2Θ testifies the X-Ray amorphous structure of PPTs and the absence of long-range order in the lattice periodicity.” on the page 4 of the subsection 3.1 Characterization of sorbents.

(2) Are lead and strontium being adsorbed simultaneously? This is not clear

Answer. Pb and Sr were not adsorbed simultaneously. We added the “(for a separate measurement of lead and strontium ions sorption)” on the page 3 in the subsection 2.2 Change of sorption rate. Thank you.

(3) "the pH of the suspension was controlled with a 150MI pH meter"  - controlled or monitored?

Answer Thank you for good advice. We wrote “monitored” instead of “controlled” on the line 84, Page 2.

(4) Some confusion in describing Figure 2 - a and b seem to be mixed in the text.

Answer. Thanks for pointing this out! Describing of figures  2a and 2b were corrected. Kindly see on the page 6.

(5) XRD looks very different to cited papers - what does “the wide reflex” mean? The assertion you make here is not supported by the data.

Answer. Thank you for this comment we added the “The broad peak, amorphous halo, in the range of 25…35 degrees of 2Θ testifies the X-Ray amorphous structure of PPTs and the absence of long-range order in the lattice periodicity.” on the page 4 in subsection 3.1.

(6) Was there any verification of the ratio of the components (using EDX?)

Answer. A typical energy-dispersive X-ray spectroscopy (EDX) analyzer attachment to an SEM EXplorer (ASPEX, USA) was used. Additional calibrations were not performed.

(7) Table 1: is there an understanding of uncertainty here. Divergence with literature values unexplained. What is calculated vs reviewed?

Answer. We added the text the text “the bond length between cation and oxygen, calculated using the density functional theory…” and “The obtained values of the bond length relatively close to literary data. The discrepancy with the literature data is due to the limited size of the calculated cluster and estimated errors of the optimization the system potential energy to the minimum” “ in the subsection  3.1 on the page 6. Thank you.

Reviewed data was taken from [Rabinovich V A and Havin Z Ya Handbook of chemistry (Leningrad: Khimiya) 1991 p 432]

(8) Results of using models are not well explained and many citations are missing. Explain the models used better and what they mean in terms of physical understanding

Answer. Thank you for the comment, We added the text “The use of the Boyd model shows complex interaction of PPTs with Pb2+ ions, and the limiting stage of external and internal diffusion is difficult to be revealed since the total sorption rate is determined by a different mechanism of diffusion transfer of metal cations.” on the page 8, then “The pseudo-first-order model describes indeed chemical sorption since most probable chemical reactions including ion exchange and covalent bonding is confirmed by good compliance of experimental data.”  on the page  9.

(9) …s throughout in numbers - are they meant to be emdashes?

Answer. To our deep regret, we feel difficulty in understanding this comment. Could the Reviewer explain what was meant? On what line?

(10) Section 3.5 needs a table

Answer. Thank you for very valuable advice. We transformed the text “PPT sorption capacity exceeds the capacity of the most known titanium sorbents, including the chitosan / TiO2 hybrid adsorbent (36.8 mg/g for Pb2+ at 60°C and pH=3...4 [43]), layered titanates Na2Ti3O7 (563.58 mg/g for Pb2+, τeq is 60…120 min [44] ………. Titanium nanotubes reaches equilibrium at 10 [52] and 30 min [53] according other studies, while the Sr2+ sorption capacity does not exceed 91.74 [52] and 98.73 mg/g [53] (pseudo-second order model).” in to the Table 3. Reported adsorption capacities towards heavy metal ions by typical adsorbents in the section 3.5 on the page 9.

(11)  Template text left in at the end of the Results and Discussion section!

Answer. Thank you, the template text “This section may be divided by subheadings. It should provide a concise and precise description of the experimental results, their interpretation as well as the experimental conclusions that can be drawn.” was deleted from the text (page 9). We are sorry.

Reviewer 3 Report

Manuscript Number : processes-690198

Title:  Wastewater treatment from lead and strontium by  potassium polytitanates: kinetic analysis and  adsorption mechanism

The present manuscript deals the waste water treatment from lead and strontium by potassium polytitanates. Although the results of the present manuscript are very promising the manuscript is not well written.  Therefore, I recommend it to be reconsidered for publication in processes after major revision. The following comments should be taken into account.

The introduction section is very extended. I suggest to minimize it. The language of the paper needs editing. (The paper is informally written. Furthermore, it should be thoroughly proof-read for grammar/syntax errors and inconsistencies) Additional references are needed in order to support the discussion of the manuscript. 4. The authors should explain the differences of their current work compared to works already published in the same field. I suggest the authors to rewrite the conclusions section. How the kinetics are affected under the presence of a magnetic , electric field ?

Author Response

Reviewer #3.

Thanks so much for reviewing our article and such valuable comments!

The present manuscript deals the waste water treatment from lead and strontium by potassium polytitanates. Although the results of the present manuscript are very promising the manuscript is not well written.  Therefore, I recommend it to be reconsidered for publication in processes after major revision. The following comments should be taken into account.

(1) The introduction section is very extended. I suggest to minimize it.

Answer. We appreciate this comment very much. When showing the environmental damage of Pb and  Sr we used references:

Iyer S.; Sengupta C.; Velumani A. Lead toxicity: An overview of prevalence in Indians. Chim. Acta 2015, 451, 161-164. Hu Q.H.; Weng J.Q.; Wang J.S. Sources of anthropogenic radionuclides in the environment: a review. Environ. Radioactiv. 2010, 101(6), 426-437. Parajuli D.; Tanaka H.; Hakuta Y.; Minami K.; Fukuda S.; Umeoka K.; Kamimura R.; Hayashi Y.; Ouchi M.; Kawamoto T. Dealing with the aftermath of Fukushima Daiichi nuclear accident: decontamination of radioactive cesium enriched ash. Sci. Tech. 2013, 47(8), 3800-3806. Lipscy P.; Kushida K.; Incerti T. The Fukushima Disaster and Japan’s Nuclear Plant Vulnerability in Comparative Perspective Sci. Tech. 2013, 47, 6082-6088. Guhathakurta H.; Kaviraj A. Heavy metal concentration in water, sediment, shrimp (Penaeus monodon) and mullet (Liza parsia) in some brackish water ponds of Sunderban, India. Pollut. Bull. 2000, 40(11), 914-920.

We agree that they are not from our field of research. We left only one:

Hu Q.H.; Weng J.Q.; Wang J.S. Sources of anthropogenic radionuclides in the environment: a review. J. Environ. Radioactiv. 2010, 101(6), 426-437.

The references were renumbered and recited in the text. Thank you.

(2) The language of the paper needs editing. (The paper is informally written. Furthermore, it should be thoroughly proof-read for grammar/syntax errors and inconsistencies)

Answer. The language of the paper was carefully edited.

All revisions were highlighted with pink color

(3) Additional references are needed in order to support the discussion of the manuscript

Answer. We agree that it is necessary to take into account all the important work in the subject area of the article. We added references in discussion and colnclusion:

Ahalya N.; Ramachandra T.V.; Kanamadi R.D. Biosorption of heavy metals. Res. J. Chem. Environ. 2008, 7(4), 71-79. Hua M.; Zhang S.; Pan B.; Zhang W.; Lv L.; Zhang Q. Heavy metal removal from water/wastewater by nanosized metal oxides: a review. Journal of hazardous materials. 2012, 211, 317-331. Rabinovich V.A.; Havin Z.Ya. Handbook of chemistry; Leningrad: Khimiya; 1991, 432.

(4) The authors should explain the differences of their current work compared to works already published in the same field

Answer. We compared the results obtained in our work on the page 9 by adding the table 3 and the text to conclusion “The sorption capacity of PPT has been calculated based on kinetic parameters study designed in accordance with pseudo-second order model and is equal to 714.3 mg/g and 344.8 mg/g for Pb2+ and Sr2+ ions, respectively.”

(5) I suggest the authors to rewrite the conclusions section

Answer. The conclusions section was improved.  We added the text “The sorption capacity of PPT has been calculated based on kinetic parameters study designed in accordance with pseudo-second order model and is equal to 714.3 mg/g and 344.8 mg/g for Pb2+ and Sr2+ ions, respectively.”, then “To sum up, the sorption capacity of potassium polytitanate sorbent obtained in this study is significantly higher than the capacity of the most ceramic materials synthesized previously [38-48]. Despite better sorption of metal cation on carbon-based sorbents [15-17], ceramic materials are more preferable because of their re-use for various practical applications.”.

Thank you for the advice

(6) How the kinetics are affected under the presence of a magnetic, electric field?

Answer. We thank for this highly valuable comment. Since the influence of magnetic and electric field on the kinetic of the sorption is very interesting question. Nevertheless, it was not a task of our research in presented paper.

Round 2

Reviewer 1 Report

Authors tried to address my previous comments and I don't have further comments. 

Author Response

Thanks again for reviewing our article!

Reviewer 2 Report

 The authors have addressed all points except point 9, which wasn't clear. I was referring to the ellipses in between numbers, for example as found in line 93: "0.2…25 kV"

Author Response

Thank you for the comment,
we specified at which voltage the presented SEM images were made (it was 15 kV), a correction was added to the text.
Some of clarifications on this comment have been added to the text, line: 172-176 and Table 3.
In addition, values in the table are rounded to significant orders of magnitude.